# Longitudinal Bone Growth Stimulating Effect of *Allium macrostemon* in Adolescent Female Rats

**DOI:** 10.3390/molecules25225449

**Published:** 2020-11-21

**Authors:** Hyung-Joong Kim, Sun Haeng Lee, Sung Hyun Lee, Jihong Lee, Hocheol Kim, Gyu Tae Chang, Donghun Lee

**Affiliations:** 1Department of Clinical Korean Medicine, Graduate School, Kyung Hee University, 26 Kyungheedae-ro, Dongdaemun-gu, Seoul 02447, Korea; siolet@naver.com (H.-J.K.); civil011@empas.com (S.H.L.); 2Department of Pediatrics of Korean Medicine, Kyung Hee University Korean Medicine Hospital, Kyung Hee University Medical Center, 23 Kyungheedae-ro, Dongdaemun-gu, Seoul 02447, Korea; 3Korea Institute of Science and Technology for Eastern Medicine (KISTEM) NeuMed Inc., 88 Imun-ro, Dongdaemun-gu, Seoul 02440, Korea; lsh@neumed.co.kr; 4Department of Pediatrics of Korean Medicine, Kyung Hee University Hospital at Gangdong, 892 Dongnam-ro, Gangdong-gu, Seoul 05278, Korea; jhcool99@hanmail.net; 5Department of Herbal Pharmacology, College of Korean Medicine, Kyung Hee University, 26 Kyungheedae-ro, Dongdaemun-gu, Seoul 02447, Korea; hckim@khu.ac.kr; 6Department of Herbal Pharmacology, College of Korean Medicine, Gachon University, 1342 Seongnamdae-ro, Sujeong-gu, Seongnam 13120, Korea

**Keywords:** *Allium macrostemon*, bone growth, insulin-like growth factor-1 (IGF-1), bone morphogenetic protein-2 (BMP-2)

## Abstract

*Allium macrostemon* (AM) may affect bone growth by regulating bone formation and resorption. To examine the effect of AM on bone growth, 48 rats were divided into four administration groups in which either distilled water, AM (100 and 300 mg/kg), or recombinant human growth hormone (rhGH; 20 μg/kg) was administered for 10 days. On day 9, all animals were intraperitoneally injected with tetracycline hydrochloride (20 mg/kg), and 48 h after the injection, the rats were sacrificed. Their tibial sections were photographed to measure bone growth. Antigen-specific immunohistochemistry was performed to detect insulin-like growth factor-1 (IGF-1) and bone morphogenetic protein-2 (BMP-2). The food intake of the AM 100 mg/kg group was higher; however, the food intake of the AM 300 mg/kg group was less than that of the control group. The rhGH and AM 100 mg/kg groups showed greater rates of bone growth (359.0 ± 23.7 and 373.1 ± 28.0 μm/day, respectively) compared with the control group. IGF-1 and BMP-2 in the AM and rhGH groups were highly expressed. Indigestion at higher doses of AM led to nonsignificant bone growth in spite of increased IGF-1 and BMP-2 expression. Therefore, a suitable amount of AM could increase bone growth.

## 1. Introduction

Longitudinal bone growth plays an important role in improving short stature [1,2]. Additionally, the bone growth is controlled by the processes of bone formation and resorption, and these processes can be detected by several bone biomarkers, such as alkaline phosphatase and hydroxyproline [3].

Some herbs, such as *Acanthopanacis senticosus* [4], *Amomum villosum* [5], *Cibotium barometz* [6], *Eucommia ulmoides* [7], *Phlomis umbrosa* [8], and *Phyllostachyos Caulis in Taeniam* [9], accelerate longitudinal bone growth in adolescent rats. Water extracts from each herb are commonly used by Korean medicine doctors (KMDs) to treat short stature in South Korea.

The *Ben Cao Gang Mu* (*Compendium of Materia Medica*) [10] is one of the 10 classical medical texts used by KMDs when they prescribe herbs. According to the *Ben Cao Gang Mu*, *Allium macrostemon* stores essential qi to the bone, harmonizes the spleen and stomach, tonifies weakness, and supports the development of a majestic physique. *A. macrostemon* may improve growth failure and short stature.

*A. macrostemon* is a perennial herb belonging to the Liliaceae family. It is generally used in Korean medicine to regulate qi. *A. macrostemon* primarily consists of alliin, methyl alliin, and scorodose. Alliin attenuates osteoclastogenesis in a dose-dependent manner [11], and allicin, which is made from alliin by alliinase, elevates low bone turnover through an increase of both bone formation and bone resorption [12]. *A. macrostemon* may affect bone growth via regulation of bone formation and resorption.

Studies show that juvenile animals supplemented with *Allium* gained more weight than control animals. Five-day-old calves given fodder containing garlic (*Allium sativum*) had a significant increase in fodder intake and weight gain [13]. Garlic-supplemented diets also significantly affected weight gain in *Clarias gariepinus* [14]. One-day-old chicks were given food containing onion (*Allium cepa* L.), and a significant increase in food intake and weight gain at 42 days of age was noted [15]. Although longitudinal bone growth was not assessed, weight gain may be related to height growth, as growth in height typically occurs with growth in weight during childhood. *A. macrostemon* may also affect food intake and weight gain and is expected to have a positive impact on growth.

Assessing growth involves measuring the long bone growth of the femur, tibia, and fibula [16]. Among the leg bones, the length of the tibia is the best indicator of stature [17]. Therefore, we used the tibia to judge the effect of *A. macrostemon* on the long bone growth.

In the study, the effect of *A. macrostemon* on tibial bone growth was verified by tetracycline fluorescent marking in the epiphyseal plate of Sprague–Dawley (SD) rats. We also investigated its effect on food intake and body weight. We used immunohistochemistry to study the expression of insulin-like growth factor-1 (IGF-1) and bone morphogenetic protein-2 (BMP-2) in the growth plate to determine a possible mechanism of *A. macrostemon*.

## 2. Results

### 2.1. Effect on Body Weight and Food Intake

Table 1 shows the body weight, and Table 2 shows the weight gain and food intake of each group. There were no statistically significant differences in body weight gain between any treatment group and the control group (*p* > 0.05). However, food intake of the *A. macrostemon* 100 mg/kg group was significantly greater than that of the control group (*p* = 0.018), while that of the *A. macrostemon* 300 mg/kg group was significantly less than that of the control group (*p* = 0.020).

### 2.2. Effect on Longitudinal Bone Growth

The longitudinal bone growths of the control group, *A. macrostemon* 100 and 300 mg/kg groups, and recombinant human growth hormone (rhGH) group were 334.8 ± 15.3, 373.1 ± 28.0 μm/day (*p* = 0.001), 342.8 ± 36.2 μm/day (*p* > 0.05), and 359.0 ± 23.7 μm/day (*p* = 0.011). The daily bone growth rates of the *A. macrostemon* 100 mg/kg and rhGH groups were significantly higher than that of the control group (Figure 1 and Figure 2).

### 2.3. Effect on IGF-1 and BMP-2 Expression

In all of the study groups, the density of IGF-1 and BMP-2 staining was higher in the cytoplasm of the proliferative and hypertrophic zones than in the resting zone of the growth plate (Figure 3 and Figure 4). The overall IGF-1 expression in the *A. macrostemon* groups was relatively higher than in the control group but lower than in the rhGH group (Table 3). The overall BMP-2 expression in the rhGH and *A. macrostemon* groups was higher than in the control group (Table 4).

## 3. Discussion

*A. macrostemon* at 100 mg/kg significantly enhanced the tibial growth rate by 11.44% compared with the control group (*p* = 0.001). This growth rate was greater than the 7.23% growth rate of the rhGH group (*p* = 0.011). An amount of 100 mg/kg of *A. macrostemon* significantly enhanced food intake by 8.82% as compared with the control group (*p* = 0.018). However, 300 mg/kg of *A. macrostemon* did not enhance the tibial growth rate as compared with the control group (*p* > 0.05). Food intake decreased in the *A. macrostemon* 300 mg/kg group (*p* = 0.020). The specific mechanism for this result is unknown. The spicy flavor of the higher dosage of *A. macrostemon* might irritate the stomach and interfere with the absorption of food or *A. macrostemon*, because spicy food can induce dyspepsia and exacerbate its symptoms [18]. These results are in line with those of previous rat studies in which raw garlic juice reduced food intake and retarded growth due to stomach injury [19]. Therefore, an appropriate amount of *A. macrostemon* should be applied to promote bone growth without stomach irritation. 

The growth plate is a cartilage between the metaphyseal and epiphyseal bone at the ends of the longitudinal bones [20]. Chondrocytes in the growth plate control longitudinal growth via proliferation, hypertrophy, apoptosis, cartilage matrix synthesis, mineralization, and vascularization [21]. The growth plate can be divided into the resting, proliferative, and hypertrophic zones according to the distinctive histological chondrocyte differentiation [22]. The resting zone is the uppermost part of the growth plate and consists of hyaline cartilage cells and a few chondrocytes. The proliferative zone is distinguished by flattened chondrocyte columns. The chondrocytes migrate to the hypertrophic zone through mitotic cell division. The cells stop dividing in the hypertrophic zone and then expand and move to the matrix, which becomes calcified. The ossification zone consists of bony spicules generated by chondrocyte apoptosis and calcium deposits [23]. 

Growth hormone (GH) and IGF-1 are crucial systemic hormones that control longitudinal bone growth during childhood [20]. GH acts upon its target tissue either directly or via insulin-like growth factors (IGFs), which are necessary components of multiple systems that regulate growth and metabolism [24]. IGF-2 is necessary for normal embryonic growth [25], whereas IGF-1 continuously regulates lifetime growth [26]. IGF-1 may intervene with the effects of GH in specific tissues or work alone in other tissues [24]. IGF-1 provokes the clonal expansion of differentiated chondrocytes in cartilage tissue [23]. Differentiated hypertrophic chondrocytes subsequently die and permit the invasion of a cell mixture that replaces the cartilage tissue with bone tissue, leading to longitudinal bone growth [27].

Bone morphogenetic proteins (BMPs) are essential to the formation of bone tissues. BMPs induce the differentiation and maturation of chondroblasts and chondrocytes [28]. Therefore, BMPs develop epiphyseal growth plates and act as therapeutic agents for bone generation [29]. BMP-2 facilitates bone formation by inhibiting osteogenesis inhibitors [30] and increases longitudinal bone growth by stimulating the proliferation and hypertrophy of growth plate chondrocytes [31].

The expression of IGF-1 and BMP-2 was elevated in the proliferative and hypertrophic zones of all rats treated with *A. macrostemon*. This suggests that the longitudinal bone growth effect of *A. macrostemon* is mediated by the local generation of IGF-1 and BMP-2 in chondrocytes. 

This experiment has some limitations. First, less than 100 mg/kg of *A. macrostemon* would have a greater bone growth-stimulating effect due to reduced stomach irritation. The concentration of *A. macrostemon* that improves bone growth should be verified with further research utilizing a diluted concentration of *A. macrostemon*. Second, serum analysis of bone turnover markers, bone growth/remodeling histology, and molecular pathway analysis are required to determine *A. macrostemon*’s exact effects on bone growth. Third, only phenotype changes were analyzed in this experiment. Further studies identifying gene expression, protein expression, and microbiota change should be performed to illuminate the potential mechanism.

## 4. Materials and Methods 

### 4.1. Plant Material and Preparation of the Extract

Dried *A. macrostemon* bulbs were obtained from Kyung Hee University Hospital at Gangdong (Seoul, Korea). They were extracted with 30% ethanol for 3 h of reflux heating. The extracted fluid was filtered with filter paper (Hyundai Micro Co., Seoul, Korea) and evaporated under reduced pressure with a rotary evaporator (Sunnileyela Co., Gyeonggi, Korea). The remaining fluid was lyophilized with a freeze dryer (Operon^TM^, Seoul, Korea), and the dark brown powder was stored at −20 °C. The freeze-dried yield of *A. macrostemon* was 47.28%.

### 4.2. Animals

There were 48 female Sprague–Dawley rats (25-day-old), weighing 60 ± 10 g each, included in this study (Samtako Co., Osan, Korea). All experimental procedures were performed according to the animal care guidelines of Kyung Hee University’s Institutional Animal Care and Use Committee (protocol number KHUASP(SE)-13-028). The animals were put in 12 cages (four rats/cage) under controlled temperature (22 ± 1 °C), relative humidity (50 ± 5%), and lightning (lights on 08:00–20:00 h) conditions, with food and water available ad libitum. The animals were evenly distributed by weight before adaptation. After 1 week of adaptation, the rats were treated for 10 days.

### 4.3. Weight and Feed Intake Check, A. macrostemon Administration, and Tetracycline Hydrochloride Injection

In previous studies involving adolescent rats, 50, 100, 200, and 300 mg/kg of herbal compounds and 100 and 500 mg/kg of herbs significantly increased longitudinal bone growth, while 30 mg/kg of herbs did not significantly enhance bone growth compared with the control group [4,5,6,7,32,33,34]. Amounts of 100 and 200 mg/kg of *A. macrostemon* had an antidepressant effect, while 400 mg/kg of *A. macrostemon* showed no effect in mice [35]. The rhGH did not enhance tibial growth or weight gain in peripubertal male rats, while an obvious growth-stimulating effect was discovered in female rats [36]. Therefore, we decided to administer 100 and 300 mg/kg of *A. macrostemon* to adolescent female rats.

The animals were randomly distributed into four groups according to administration. Distilled water (DW) (12 rats) and *A. macrostemon* (100 mg/kg in 12 rats and 300 mg/kg in 12 rats) were given via a 10.0 mL/kg gavage twice daily (at 9:00 and 21:00 h). The rhGH (20 μg/kg in 12 rats) (LG Life Sciences, Seoul, Korea) was subcutaneously injected once daily at 9:00 h at a dosage of 1.0 mL/kg. The body weight of every animal and the food intake per cage were checked daily before administration. We maintained the treatment for 10 consecutive days. On day 9, all the animals were intraperitoneally injected with tetracycline hydrochloride (20 mg/kg, Sigma, St. Louis, MO, USA) in a 5.0 mL/kg saline for fluorescent marking under ultraviolet illumination. The animals were sacrificed with cervical dislocation 48 h after the injection. We dissected the tibias of each animal free of the soft tissue and fixed the bones in a 4% paraformaldehyde for 24 h.

### 4.4. Tissue Preparation and Detection of Longitudinal Bone Growth

Fixed tibias were decalcified by immersing them in a 50 mM ethylene diamine tetraacetate solution (Sigma) for 48 h. They were dehydrated by immersion in a 30% sucrose solution for 24 h. Each 40 μm thick dehydrated bone was sectioned longitudinally with a sliding microtome (Leica, Berlin, Germany). The bone sections were mounted onto gelatinized glass slides and viewed by fluorescence microscopy (Olympus, Tokyo, Japan). The longitudinal bone length between the fluorescent line and the epiphyseal end line of the growth plate was measured by two administration-blinded assessors using ImageJ version 1.43u (National Institutes of Health, Bethesda, MD, USA). Half of the average bone length at three different sample locations was the daily growth rate.

### 4.5. Measurement of IGF-1 and BMP-2 in the Growth Plates

Tissue sections were washed twice in a 0.1 M phosphate-buffered saline (PBS) for 15 min and incubated in a 0.1 M PBS/1% bovine serum albumin (BSA, Sigma)/1% Triton X-100 (Sigma) solution for 10 min. The tissues were then washed twice with a solution of 0.1 M PBS/0.5% BSA for 15 min. The sections were incubated with rabbit IGF-1 primary antibody and goat BMP-2 primary antibody (1:200, Santa Cruz Biotechnology, CA, USA) overnight at room temperature in a humid chamber. After 24 h, sections were washed twice with 0.5% BSA/0.1 M PBS, and then incubated with the biotinylated anti-rabbit secondary antibody (1:200, Jackson ImmunoResearch Laboratories, West Grove, PA, USA) and biotinylated anti-goat secondary antibody (1:200, Vector Laboratories, Burlingame, CA, USA) for 60 min, respectively. After washing twice with 0.5% BSA/0.1 M PBS for 15 min, the sections were incubated with an avidin–biotin–peroxidase complex (1:100, Vectastain ABC Kit, Vector Laboratories, Burlingame, CA, USA) for 1 h at room temperature. The sections were stained with 3,3’-diaminobenzidine tetrahydrochloride dehydrate after washing twice with a 0.1 M phosphate buffer (PB). The staining was stopped with PB washing, and the sections were mounted onto slides and twice dehydrated with solutions of DW, 50%, 75%, 95%, and 100% ethanol, and xylene in that order. The slides were enveloped with cover slides and a Permount medium solution (Fisher Scientific, city, NH, USA). The sections were photographed with a microscope.

### 4.6. Statistical Analysis

Results were presented as the mean ± standard deviation. Statistically significant values were analyzed using ANOVA among four groups, Tukey’s multiple comparison test for the post hoc analysis, and Student’s *t*-test after the normality test for comparisons between each treatment and the DW groups. Data were analyzed using IBM SPSS 19.0 (IBM Corp., Armonk, NY, USA). *p*-Values < 0.05 were statistically significant.

## 5. Conclusions

Both 100 and 300 mg/kg doses of *A. macrostemon* enhanced IGF-I and BMP-2 generation in the growth plates. However, only 100 mg/kg dosage of *A. macrostemon* demonstrated a significant longitudinal bone growth effect. While 100 mg/kg of *A. macrostemon* significantly increase food intake, 300 mg/kg of *A. macrostemon* significantly decrease food intake. Indigestion associated with higher doses of *A. macrostemon* may have led to nonsignificant longitudinal bone growth despite increased expression of IGF-1 and BMP-2. Therefore, a suitable amount of *A. Macrostemon* could increase the rate of longitudinal bone growth. Further research is needed to clarify the effective dosage.

## Figures and Tables

**Figure 1 molecules-25-05449-f001:**
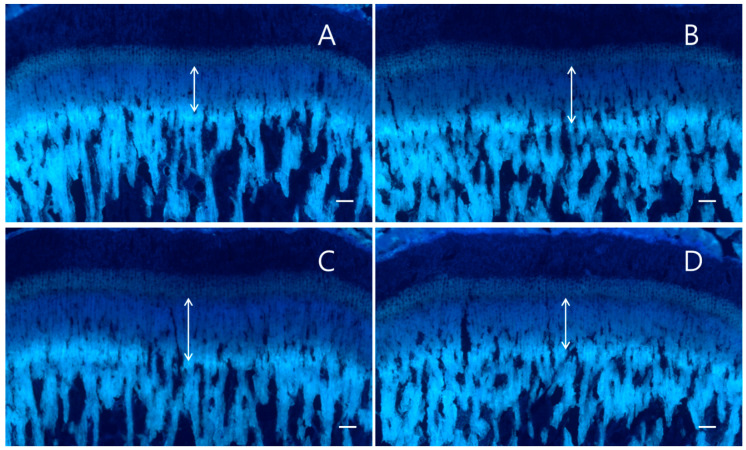
Fluorescence photomicrograph of a longitudinal section of the proximal tibia. The fluorescent line corresponds to the tetracycline hydrochloride (20 mg/kg) injection. The arrow indicates the length of growth. (**A**) Control (distilled water) group, (**B**) recombinant human growth hormone (20 μg/kg, s.c.) group, (**C**) *A. macrostemon* 100 mg/kg group, (**D**) *A. macrostemon* 300 mg/kg group. Scale bar = 200 μm.

**Figure 2 molecules-25-05449-f002:**
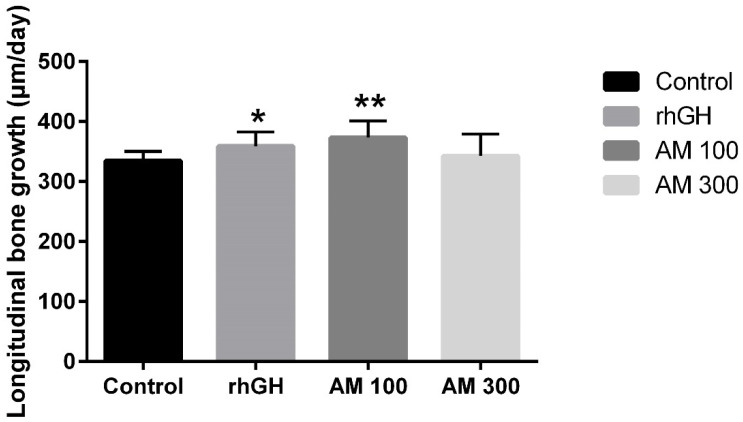
The longitudinal bone growth rate. Control: distilled water (10 mL/kg, p.o.) group; rhGH: recombinant human growth hormone (20 μg/kg, s.c.) group; AM 100: *A. macrostemon* (100 mg/kg, p.o.) group; AM 300: *A. macrostemon* (300 mg/kg, p.o.) group. Each value is the mean ± standard deviation of 12 rats. Statistical significance was determined using a *t*-test: * *p* < 0.05, ** *p* < 0.01, compared with control.

**Figure 3 molecules-25-05449-f003:**
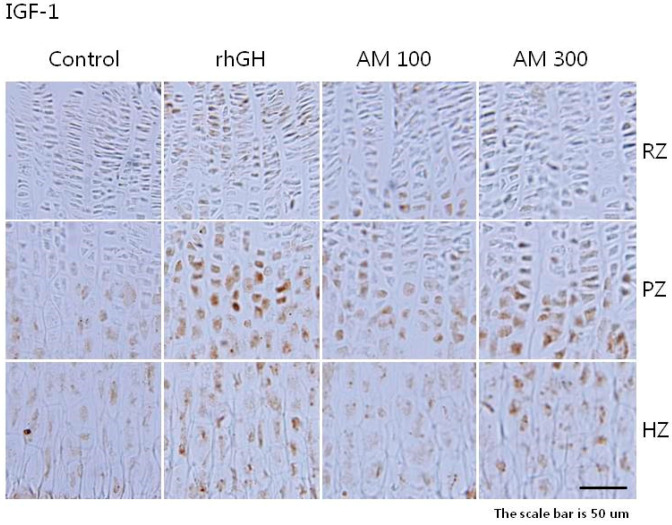
Immunohistochemical localization of insulin-like growth factor-1 in the growth plate. Control: distilled water (10 mL/kg, p.o.) group; rhGH: recombinant human growth hormone (20 μg/kg, s.c.) group; AM 100: *A. macrostemon* (100 mg/kg, p.o.) group; AM 300: *A. macrostemon* (300 mg/kg, p.o.) group. RZ: resting zone; PZ: proliferative zone; HZ: hypertrophic zone. Scale bar = 50 μm.

**Figure 4 molecules-25-05449-f004:**
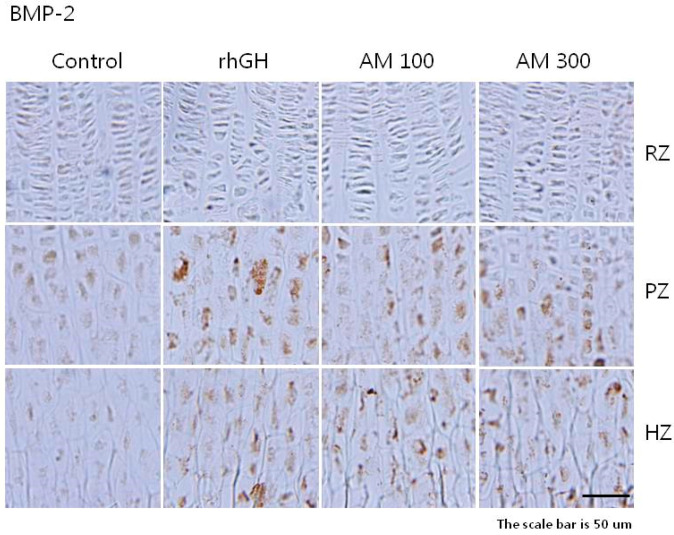
Immunohistochemical localization of bone morphogenetic protein-2 in the growth plate. Control: distilled water (10 mL/kg, p.o.) group; rhGH: recombinant human growth hormone (20 μg/kg, s.c.) group; AM 100: *A. macrostemon* (100 mg/kg, p.o.) group; AM 300: *A. macrostemon* (300 mg/kg, p.o.) group. RZ: resting zone; PZ: proliferative zone; HZ: hypertrophic zone. Scale bar = 50 μm.

**Table 1 molecules-25-05449-t001:** Daily body weight of female adolescent rats.

Day	Control	rhGH	AM 100	AM 300	*p*-Value
1	106.8 ± 6.2	106.5 ± 4.4	107.9 ± 5.4	106.5 ± 3.4	0.892
2	110.6 ± 7.2	109.7 ± 5.2	112.4 ± 5.0	109.2 ± 3.4	0.482
3	117.1 ± 7.3	117.3 ± 5.8	119.7 ± 5.4	115.8 ± 3.2	0.414
4	121.9 ± 7.8	122.0 ± 5.9	124.8 ± 5.2	120.3 ± 3.9	0.314
5	126.5 ± 7.9	126.9 ± 6.0	130.6 ± 6.0	124.6 ± 3.0	0.111
6	131.5 ± 8.2	131.2 ± 6.2	134.8 ± 6.0	128.5 ± 3.0	0.111
7	137.0 ± 7.9	137.2 ± 7.4	140.7 ± 6.4	133.8 ± 3.8	0.101
8	139.8 ± 8.4	140.4 ± 7.2	144.3 ± 6.1 ^#^	135.9 ± 4.0 ^#^	0.031 *
9	143.3 ± 8.0	144.0 ± 7.1	149.2 ± 7.6 ^#^	140.6 ± 4.6 ^#^	0.033 *
10	145.3 ± 7.4	145.7 ± 8.9	150.9 ± 8.9 ^#^	140.1 ± 6.7 ^#^	0.019 *
11	147.6 ± 8.1	146.8 ± 9.3	152.1 ± 9.3 ^#^	142.8 ± 6.3 ^#^	0.068

Control: distilled water (10 mL/kg, p.o.) group; rhGH: recombinant human growth hormone (20 μg/kg, s.c.) group; AM 100: *A. macrostemon* (100 mg/kg, p.o.) group; AM 300: *A. macrostemon* (300 mg/kg, p.o.) group. Each value is the mean ± standard deviation of 12 rats. Statistical significance was determined using ANOVA: * *p* < 0.05. ^#^ indicates statistical significance based on Tukey’s multiple comparison.

**Table 2 molecules-25-05449-t002:** Body weight and daily food intake gains (g) over 10 days in female adolescent rats.

	Control	rhGH	AM 100	AM 300	*p*-Value
Body weight gain (%/rat)	38.26 ± 3.99	37.88 ± 6.86	41.13 ± 7.69	34.20 ± 8.07	0.118
Daily food intake gain (g/cage)	597.3 ± 7.7 ^#^	613.5 ± 19.6 ^#^	650.0 ± 22.1 ^#^	565.3 ± 12.6 ^#^	0.002 **

Control: distilled water (10 mL/kg, p.o.) group; rhGH: recombinant human growth hormone (20 μg/kg, s.c.) group; AM 100: *A. macrostemon* (100 mg/kg, p.o.) group; AM 300: *A. macrostemon* (300 mg/kg, p.o.) group. Each value is the mean ± standard deviation of 12 rats or three cages. Statistical significance was determined using ANOVA: ** *p* < 0.01. ^#^ indicates statistical significance based on Tukey’s multiple comparison.

**Table 3 molecules-25-05449-t003:** Immunohistochemical localization of IGF-1 on the proximal tibial growth plate in rats.

	Control	rhGH	AM 100	AM 300	*p*-Value
Overall expressions (n)	30.7 ± 2.9 ^#^	92.7 ± 13.8 ^#^	64.7 ± 8.4	66.0 ± 4.4	< 0.001 ***
Resting zone	4.3 ± 0.6 ^#^	17.7 ± 4.5 ^#^	11.3 ± 0.6 ^#^	6.3 ± 1.5	0.001 **
Proliferative zone	12.0 ± 1.7 ^#^	50.0 ± 5.2 ^#^	32.3 ± 4.5	34.3 ± 5.1	< 0.001 ***
Hypertrophic zone	14.3 ± 2.1 ^#^	25.0 ± 4.4 ^#^	21.3 ± 4.5	25.3 ± 0.6 ^#^	0.012 *

Control: distilled water (10 mL/kg, p.o.) group; rhGH: recombinant human growth hormone (20 μg/kg, s.c.) group; AM 100: *A. macrostemon* (100 mg/kg, p.o.) group; AM 300: *A. macrostemon* (300 mg/kg, p.o.) group. Each value is the mean ± standard deviation of 12 rats or three cages. Statistical significance was determined using ANOVA: * *p* < 0.05, ** *p* < 0.01, *** *p* < 0.001. ^#^ indicates statistical significance based on Tukey’s multiple comparison.

**Table 4 molecules-25-05449-t004:** Immunohistochemical localization of BMP-2 on the proximal tibial growth plate in rats.

	Control	rhGH	AM 100	AM 300	*p*-Value
Overall expressions (n)	24.0 ± 3.6 ^#^	52.3 ± 5.5	42.0 ± 6.6	44.7 ± 4.9	0.001 **
Resting zone	2.7 ± 3.1	5.7 ± 5.5	3.7 ± 4.0	3.7 ± 3.5	0.840
Proliferative zone	12.3 ± 3.1 ^#^	27.7 ± 3.5	22.0 ± 3.6	24.3 ± 3.5	0.003 **
Hypertrophic zone	9.0 ± 1.0 ^#^	19.0 ± 2.0	16.3 ± 2.1	16.7 ± 3.1	0.002 **

Control: distilled water (10 mL/kg, p.o.) group; rhGH: recombinant human growth hormone (20 μg/kg, s.c.) group; AM 100: *A. macrostemon* (100 mg/kg, p.o.) group; AM 300: *A. macrostemon* (300 mg/kg, p.o.) group. Each value is the mean ± standard deviation of 12 rats or three cages. Statistical significance was determined using ANOVA: ** *p* < 0.01. ^#^ indicates statistical significance based on Tukey’s multiple comparison.

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
