# Peer review of "Longitudinal Bone Growth Stimulating Effect of Allium macrostemon in Adolescent Female Rats"

_molecules, 2020, doi:10.3390/molecules25225449_

Round 1
Reviewer 1 Report
The present study aims to illuminate the effect of Allium Macrostemon on the Longitudinal Bone Growth in mice. Although some significant effects on Allium macrostemon growth were identified, there were several limitation in the present study. Hence, the manuscript could not be accepted as the present form. My main concerns were listed as below.
1) It is rather difficult to read and understand the English of the manuscript. The English of the manuscript needs to be improved with the help from a native English speaker.
2) Student’s t-test was not suitable for the present study, all data should be re-analyzed.
3) For Figure 2, 3, and 4, the data should be measured with replication, and then draw a conclusion with further statistical analysis.
4) All data presented in this manuscript were all phenotype changes, further study should be performed to illuminate the potential mechanism. Gene expression, protein expression, and microbiota changes.
Author Response
Comment 1:
It is rather difficult to read and understand the English of the manuscript. The English of the manuscript needs to be improved with the help from a native English speaker.
Response 1:
Thank you for your comment. We took English correction by editing service of ESSAYREVIEW. We submitted editorial certificate.
Comment 2:
Student’s t-test was not suitable for the present study, all data should be re-analyzed.
Response 2:
Thank you for your comment. We analyzed the data using ANOVA among four groups. (see pages 8 in the revised manuscript, marked in red).
“Statistically significant values were analyzed using ANOVA among four groups, Tukey’s multiple comparison test for the post-hoc analysis, and Student’s t-test after the normality test for comparisons between each treatment and DW groups.”
Comment 3:
For Figure 2, 3, and 4, the data should be measured with replication, and then draw a conclusion with further statistical analysis.
Response 3:
Thank you for your comment. We measured the data by two administration-blinded assessors (H.-J.K. and S.Hy.L) and then draw a conclusion with ANOVA. (see pages 7 in the revised manuscript, marked in red).
“The longitudinal bone length between the fluorescent line and the epiphyseal end line of the growth plate was measured by two administration-blinded assessors using Image J version 1.43u (National Institutes of Health, USA).”
Comment 4:
All data presented in this manuscript were all phenotype changes, further study should be performed to illuminate the potential mechanism. Gene expression, protein expression, and microbiota changes.
Response 4:
Thank you for your comment. As you have advised, we commented as the limitation of the experiments (see pages 6 in the revised manuscript, marked in red).
“Third, only phenotype changes were analyzed in this experiment. Further studies identifying gene expression, protein expression, and microbiota change should be performed to illuminate the potential mechanism.”
Reviewer 2 Report
The authors have examined AM on bone growth with mouse model. The tibial sections of all mice have been photographed to measure bone growth. The authors have also detected IGF-1 and BMP-2 using antigen-specific immunohistochemistry. The rhGH and AM 100 mg/kg groups have showed a greater rate of bone growth compared to the control group. Moreover, IGF-1 and BMP-2 in the AM and rhGH 31 groups have highly expressed. In this wok, the authors have demonstrated that a suitable amount of AM could increase bone growth. Overall, this work can inspire more herbal investigations for bone growth. The paper is basically well done and well written. Therefore, I would like to recommend this work to publish in Molecules. Herein, I only have one suggestion. In the introduction, for the first sentence “Longitudinal bone growth plays an important role in improving short stature”, more references should be cited to broaden the introduction more completely. For example, several studies have reported the bone biomarker for the clinical applications. (Biomarker Research (2017) 5:18)
Author Response
Comment 1:
Herein, I only have one suggestion. In the introduction, for the first sentence “Longitudinal bone growth plays an important role in improving short stature”, more references should be cited to broaden the introduction more completely. For example, several studies have reported the bone biomarker for the clinical applications. (Biomarker Research (2017) 5:18)
Response 1:
Thank you for your comment. We revised the introduction according to your comment (see pages 1 in the revised manuscript, marked in red).
“And the bone growth is controlled by the process of bone formation and resorption, and these processes can be detected by several bone biomarkers such as alkaline phosphatase and hydroxyproline [1].”
Reviewer 3 Report
In the study, the authors tried to test the effects of Allium macrostemon on tibial bone growth in adolescent female Sprague-Dawley (SD) rats. The authors suggested that the longitudinal bone growth stimulating effect of A. macrostemon was mediated by the local generation of IGF-1 and BMP-2 in chondrocytes.
Comments
The reviewer has some concerns as follows:
- The stimulating effects of longitudinal bone growth rate by both rhGH and AM treatment are really limited (Figure 2). Is the duration of treatment (10 days) proper?
- The authors mentioned that statistical analysis was analyzed using Student’s t-test after the normality test for comparisons between groups. It is confusing for “after the normality test for comparisons between groups”. The Student’s t-test for four experimental groups in this study is improper. The authors should revise the statistical analysis.
- In Table 1, please provide the data for body weights of all groups, but not just the gain values.
- In Figure 2, please provide the number of tibia (n number) in each group for analysis.
- In Figures 3 and 4, the quantification for immunohistochemical localization of IGF-1 and BMP-2 is recommended to support the data of longitudinal bone growth rate.
- A conclusion section is recommended in this manuscript.
Author Response
Comment 1:
The stimulating effects of longitudinal bone growth rate by both rhGH and AM treatment are really limited (Figure 2). Is the duration of treatment (10 days) proper?
Response 1:
Thank you for your comment. Rats become sexually mature at about 6 weeks of age. In this study, we prepared the 25-day-old rats and adapted for 7 days. After the 10 days of treatment (at about 6 weeks of age), we can check the growth of adolescence which is characterized by sexually maturation.
Comment 2:
The authors mentioned that statistical analysis was analyzed using Student’s t-test after the normality test for comparisons between groups. It is confusing for “after the normality test for comparisons between groups”. The Student’s t-test for four experimental groups in this study is improper. The authors should revise the statistical analysis.
Response 2:
Thank you for your comment. We used the Student’s t-test between two groups (see pages 1 in the revised manuscript, marked in red).
“Statistically significant values were analyzed using ANOVA among four groups, Tukey’s multiple comparison test for the post-hoc analysis, and Student’s t-test after the normality test for comparisons between each treatment and DW groups..”
Comment 3:
In Table 1, please provide the data for body weights of all groups, but not just the gain values.
Response 3:
Thank you for your comment. We added the table for body weights of all groups (see table 1).
Comment 4:
In Figure 2, please provide the number of tibia (n number) in each group for analysis.
Response 4:
Thank you for your comment. We revised the legend of figure 2 (see the legend of figure 2).
“Each value is the mean ± standard deviation of 12 rats.”
Comment 5:
In Figures 3 and 4, the quantification for immunohistochemical localization of IGF-1 and BMP-2 is recommended to support the data of longitudinal bone growth rate.
Response 5:
Thank you for your comment. We quantified the immunohistochemical localization of IGF-1 and BMP-2 according to your comment (see table 3 and 4).
“Overall IGF-1 expression in the A. macrostemon groups were relatively higher than in the control group, but lower than in the rhGH group (Table 3). Overall BMP-2 expression in the rhGH and A. macrostemon groups were higher than in the control group (Table 4).”
Comment 6:
A conclusion section is recommended in this manuscript.
Response 6:
Thank you for your comment. We made a conclusion section.
Round 2
Reviewer 1 Report
The authors have improved their manuscript according to my comments. I think the manuscript could be accepted as the present form.
Author Response
The authors have improved their manuscript according to my comments. I think the manuscript could be accepted as the present form.
-> Thanks for your kind and rapid response.
Reviewer 3 Report
No further comments for this revised manuscript. It can be accepted to publish in this Journal.
Author Response
No further comments for this revised manuscript. It can be accepted to publish in this Journal.
-> Thanks for your kind and rapid response.